# Global impact of tobacco control policies on smokeless tobacco use: a systematic review protocol

Monika Arora [1,2] Aastha Chugh [1] Neha Jain [2] Masuma Mishu,[3]
Melanie Boeckmann [4] Suranji Dahanayake,[3,5] Jappe Eckhardt,[6]
Sarah Forberger [7] Rumana Huque,[8] Mona Kanaan,[3] Zohaib Khan [9]
Ravi Mehrotra,[10] Muhammad Aziz Rahman,[11,12] Anne Readshaw,[3] Aziz Sheikh,[13]
Kamran Siddiqi [3] Aishwarya Vidyasagaran,[3] Omara Dogar[3,13]

► Prepublication history and additional materials for this paper is available online. To view these files, please visit the journal online (http://dx.doi.org/10.1136/bmjopen-2020-042860).

For numbered affiliations see end of article.

**Correspondence to**
Professor Melanie Boeckmann;
boeckmannmelanie@gmail.com

## ABSTRACT

**Introduction** Smokeless tobacco (ST) was consumed by 356 million people globally in 2017. Recent evidence shows that ST consumption is responsible for an estimated 652 494 all-cause deaths across the globe annually. The WHO Framework Convention on Tobacco Control (FCTC) was negotiated in 2003 and ratified in 2005 to implement effective tobacco control measures. While the policy measures enacted through various tobacco control laws have been effective in reducing the incidence and prevalence of smoking, the impact of ST-related policies (within WHO FCTC and beyond) on ST use is under-researched and not collated.

**Methods and analysis** A systematic review will be conducted to collate all available ST-related policies implemented across various countries and assess their impact on ST use. The following databases will be searched: Medline, EMBASE, PsycINFO, Cumulative Index to Nursing and Allied Health Literature, Scopus, EconLit, ISI Web of Science, Cochrane Library (CENTRAL), African Index Medicus, LILACS, Scientific Electronic Library Online, Index Medicus for the Eastern Mediterranean Region, Index Medicus for South-East Asia Region, Western Pacific Region Index Medicus and WHO Library Database, as well as Google search engine and country-specific government websites. All ST-related policy documents (FCTC and non-FCTC) will be included. Results will be limited to literature published since 2005 in English and regional languages (Bengali, Hindi and Urdu). Two reviewers will independently employ two-stage screening to determine inclusion. The Effective Public Health Practice Project's 'Quality Assessment Tool for Quantitative Studies' will be used to record ratings of quality and risk of bias among studies selected for inclusion. Data will be extracted using a standardised form. Meta-analysis and narrative synthesis will be used.

**Ethics and dissemination** Permission for ethics exemption of the review was obtained from the Centre for Chronic Disease Control's Institutional Ethics Committee, India (CCDC_IEC_06_2020; dated 16 April 2020). The results will be disseminated through publications in a peer-reviewed journal and will be presented in national and international conferences.

**PROSPERO registration number** CRD42020191946.

### Strengths and limitations of this study

► This is a protocol for a systematic review that is a comprehensive global review of all available Smokeless tobacco (ST)-related policies.

► The review will include all available ST-related policies (Framework Convention on Tobacco Control, FCTC and non-FCTC policies). It will also include ST policies notified/enforced by various government bodies such as Ministry of Health, Ministry of Finance, Ministry of Commerce.

► This review is the first attempt to search for ST-related policies globally and assess their impact systematically across academic databases and country-specific government websites.

► Studies and policy documents in English and regional languages for which expertise is available in the team (Bengali, Hindi and Urdu) will be included in the review. Due to limited resources, documents in other languages will not be included.

## INTRODUCTION

Tobacco products are broadly classified into two categories—smoked and smokeless tobacco (ST). The WHO's Framework Convention on Tobacco Control (WHO FCTC) defines ST as 'tobacco that is consumed in un-burnt form either orally or nasally'.[1] A wide range of ST products are manufactured and consumed worldwide, some of which are snus, pan masala, gutkha, mawa, khaini, zarda, mishri, dry snuff, moist snuff. ST was consumed by 356 million people globally in 2017.[2] Cumulatively, it is estimated that, globally, a total of 652 494 people die from ST attributed diseases annually.[3] It is used in about 116 countries around the world,[4] and a vast majority of users reside in South and South-East Asia, which bears more than 85% of the burden of diseases due to ST use.[5] The practice of ST use is culturally influenced and widespread in this region, where

one-third of tobacco is consumed in the smokeless form.[6] This widespread use of ST products is a cause for alarm, considering that they have been known to contain more than 30 carcinogens.[6] The Effect of Potentially Modifiable Risk Factors Associated with the Myocardial Infarction (INTERHEART) study, a large case–control study published in 2006 and undertaken in 52 countries, found that 4.7 million Disability Adjusted Life Years (DALYs) were lost and 204 309 people died from coronary heart diseases caused by ST consumption.[7] ST use leads to oral submucous fibrosis and cancers of the oral cavity.[8] Pregnant women using ST have shown a threefold increased risk of stillbirth and twofold to threefold increased risk of having low-birthweight babies.[9–11]

The WHO FCTC was negotiated in 2003, and ratified in 2005, with the aim of providing guidelines to implement effective tobacco control measures.[1] Currently, there are 183 country-level signatories to this treaty. Though the menace of ST use is considered to be relevant to fewer countries and regions than smoking, it is evident that there has been an increase in the prevalence of ST use, even in economically developed countries.[12] However, the situation is worse in low-income and middle-income countries (LMICs), such as India, where the prevalence of ST use is higher than smoking.[13] While 10.7% of all the adults in India are smokers, 21.4% are current ST users.[13] Similarly in Bangladesh, while 18% of all adults are smokers, 20.6% are ST users.[14]

Furthermore, while the policies described under FCTC have been successful in decreasing the incidence and prevalence of smoking, there is insufficient evidence on the impact of the same on ST use.[2] For example, taxation and smoke-free policies (as described in Article 6 and 8 of the FCTC) have been successful in reducing the prevalence of smoking in countries such as Greece.[15] However, the impact of these policies on ST use is not widely known, nor have they been systematically collated. This could be attributed to the fact that ST has not been given the same importance as smoking while implementing these policies. In addition, these policies have often been based on evidence gathered from research on cigarettes conducted in high-income countries, and hence the translation of the same policies for the control of ST in LMICs has not been contextualised.[16] For instance, mass media campaigns for raising awareness on ST have been highlighted by very few countries, and sale of ST products to minors is illegal only in a handful of the countries.[2 17] In addition, secondhand impact of ST use (such as diseases linked to spitting) and its adverse environmental impacts due to disposal of small sachets of ST products) do not invoke any public health response. The burden of ST is now being realised in some developed countries too. For example, Australia recently imposed a ban on import, production, sale, distribution and advertisement of ST products, including chewing tobacco, oral snuff, tobacco paste and powder.[18] On the other hand, in the USA, the Food and Drug Administration now considers ST as a modified risk tobacco product, under their Family Smoking Prevention and Tobacco

Control Act 2009.[12] This change in policy scenario has been neither persistent nor uniform across the world. Countries like India and Sri Lanka have gone beyond FCTC to include the prohibition of some (eg, gutka) or all ST products.[19] The impact of FCTC policies, as well as country-specific tobacco control laws, on the consumption of ST is under-researched and not collated.

## AIMS AND OBJECTIVES

The study aims to undertake a systematic review to collate all policies implemented across countries globally relevant to the control of ST products and assess their impact on ST use.

Specific review questions (RQs) include:

### Review question 1

What are the existing policies and legislation (FCTC and non-FCTC) related to the control of ST products across various countries globally?

### Review question 2

What is the impact of these policies on controlling ST use, including its uptake, cessation and reduction?

## METHODOLOGY

The study is a systematic review which will consider all relevant studies from 2005 onwards (after the WHO FCTC came into force) without any geographical restriction. Articles in English and for which linguistic expertise is available within the team (Bengali, Hindi and Urdu) will be included. Our study will include published articles (including commentaries and editorials) searched through scientific databases. Policy documents (policies/laws/notifications/Acts related to ST) identified through grey literature searches will also be included in the review. Systematic reviews will be excluded because we expect to cover most of the relevant literature through our searches and a review of systematic reviews will be beyond the scope of the study. Studies reporting at least one of the outcomes (detailed below) in relation to at least one policy of interest will be eligible for inclusion. We will identify studies using keywords such as 'ST', 'public policies', with synonyms, and based on the PICOS criteria outlined below.

### Population

We will include studies on ST users (not e-cigarettes and other nicotine products-Electronic Nicotine Delivery Systems) of all age groups.

### Interventions

Policy documents (FCTC and non-FCTC) within health and other sectors (commerce, finance, etc) that describe the regulation or prohibition of ST products directly or indirectly will be considered eligible for RQ1. In addition, only those policies which have been implemented/enforced by the government will be included in the study for analysis.

Policies which are at the planning stage (not implemented/ enforced) until June 2020 will not be included. Specifically, we will include policies/legislations notified and enforced by governmental bodies such as Ministry of Health, Ministry of Finance, at the country level. Documents related to the following policies will be included in the review:

FCTC policies:
► Pricing and taxation (Article 6).
► Product regulation (Article 9 and 10).
► Packaging and health warnings (Article 11).
► Education, communication, training and public awareness (Article 12).
► Advertisement, promotion and sponsorship bans (Article 13).
► Cessation (Article 14).
► Illicit trade (Article 15).
► Sales to and by minors (Article 16).
  Non-FCTC policies:
► Complete or partial ban on ST products (eg, gutkha ban).
► Import ban.
► Other policies mentioned to control ST (agriculture, environment etc).

For RQ2, studies that report on the impact of the above-mentioned policies on ST related outcomes (see below) will be included.

We will exclude studies primarily looking at policies on harm reduction or switching between tobacco products. However, if the primary focus of studies is ST cessation then these studies will be considered eligible.

### Comparators
Absence of intervention or usual care will be considered as comparators.

### Outcomes
The outcomes to be included in the review will depend on the research questions. While RQ1 will be descriptive and will collate all policy measures undertaken in various countries, RQ2 will focus on studies reporting any of the following outcome measures in relation to specific policies (FCTC and non-FCTC).
1. Primary outcomes
   – Percentage change in prevalence of ST use.
   – Rates of quitting ST (proportion of those who stopped out of total intervened).
   – Percentage change in ST initiation (from never used to ever used).
   – Change in health outcomes for cancers and cardio-vascular diseases attributable to ST (ST-related disease burden is substantial for these conditions, as noted in a recent study).[20]
2. Intermediate outcomes
   – ST availability,affordability and accessibility.
   – Change in knowledge, attitude, perception and norms related to ST use.
   – Change in levels of chemical constituents.
   – Noticing health warnings/advertisements.

– Recall of awareness campaigns.
3. Reported unintended effects

Policy documents, studies, editorials and commentaries, observational (cross-sectional/cohort/case–control) and intervention studies (randomised controlled trials/quasi experimental studies/economic evaluations) will be included in the review.

### DATABASES
1. For published studies: The following global and re-gional databases will be searched to identify relevant published studies: Medline, EMBASE, PsycINFO, Cumulative Index to Nursing and Allied Health Literature, Scopus, EconLit, ISI Web of Science, Cochrane Library (CENTRAL), African Index Medicus (AIM), LILACS, Scientific Electronic Library Online, Global Medicus Index (which includes Index Medicus for the Eastern Mediterranean Region, Index Medicus for South-East Asia Region, AIM and Western Pacific Region Index Medicus, and WHO Library Database.
2. For grey literature: Grey literature search will be conducted on two different platforms as follows:
   a. Google search engine (first 100 hits) and target web-sites known within the research team such as WHO, Campaign for Tobacco Free Kids (CTFK), Interna-tional clinical trials registry platform (WHO), EU Clinical Trials Register, ISRCTN, ProQuest or those identified through Google search.
   b. Country-specific government websites will be searched to collate policy documents (policies/ laws/notifications/Acts) related to ST (FCTC and non-FCTC) in four South Asian countries (Bangla-desh, India, Pakistan and Sri Lanka). The ministry websites searching will be restricted to South Asian countries only, given the time frame and scope of the current study as well as due to language limi-tations. We will search websites of the Ministry of Health, Ministry of Commerce, Ministry of Finance, Ministry of Environment and governmental bodies that might be relevant to either of the four coun-tries.

### SEARCH STRATEGY
1. For published studies: The literature search will be conducted with key terms such as 'ST', 'policies', 'legis-lation', with their synonyms (search string attached as online supplemental file 1). We will identify all ST con-trol policies from around the world to address RQ1. For RQ2, from the same search, we will include studies that have estimated impact on ST use.
   References from other systematic reviews and chap-ters/textbooks will be assessed to identify potential studies, in addition to hand searching reference lists and citations of included studies.
2. For grey Literature: Keywords including 'ST' and 'pub-lic policy' along with their relevant synonyms will be

used to run the grey literature search. Key terms will be in alignment with the search terms of the literature review, however, will be adapted to the specific contexts of the search engines (google search engine or ministry website search engine).

a. Google search engine: Google search engine will be used to search ST related documents published on the Internet. During our search, the first 100 hits or hits for results going back till the year 2005 (whichever comes first) will be screened for inclusion in the study. The results will be assessed using the title and the short description underneath, and all relevant records will be bookmarked in the browser and also entered into an Excel spreadsheet. Each entry will be made under the heading which has been titled as per the search strategy used. If the results include documents from India, Pakistan, Sri Lanka or Bangladesh, they will be saved in a separate excel spreadsheet to check for duplicates from the Ministry website searches. Newspaper articles will only be screened for identifying ST policies or their impact evaluation. Further, potential websites (such as WHO, CTFK) will be identified by Google search to identify ST policies or impact assessment studies related to ST policies. The dates of searching these websites will be documented in an excel spreadsheet. For the websites with a search engine, keywords will be used for searching the documents. For the websites without a search engine, hand searching will be conducted to identify relevant policies or impact assessment studies. The name, year and URL of these individual records within the websites would also be documented in the excel spreadsheet, as a subheading of the main website. All relevant entries that satisfy the inclusion criteria will then be screened.

b. Country-specific government websites: A search using keywords, 'ST' and 'public policy', with country-specific synonyms will be conducted on government websites across four SEAR countries (Bangladesh, India, Pakistan and Sri Lanka). All policy documents (policies, laws, legislations, acts and notifications) related to ST (FCTC and non-FCTC) will be searched on relevant ministry websites. Country-level experts on the team who have experience in ST research will be consulted to select relevant policy documents from the searches. Only national-level documents will be selected for this analysis as these policy guidelines prescribe the characteristics for subnational policies. The title and summary of the policy documents will be screened. Details of potentially relevant records from all four countries will be added into an excel spreadsheet for full-text screening. Relevant policy documents after full-text screening and deduplication with google search results, will be included in the review.

To avoid duplication of the work and ensure comparable approaches, regular meetings to discuss search strategies among the team will be held.

## STUDY RECORDS
### Data management
Records from scientific databases will be imported into CADIMA (https://www.cadima.info/), which is an open access online tool for conducting systematic reviews[21]; and deduplicated.

### Study selection
All the studies identified through database searching and policy documents identified through grey literature searches will be assessed by two reviewers independently for inclusion in the review based on predefined inclusion criteria (stage 1: titles and/or abstracts, stage 2: full text). Any disagreement will be resolved through mutual discussion and if needed, through a third reviewer.

### Data extraction
Data extraction procedures will follow Preferred Reporting Items for Systematic Reviews and Meta-Analyses (PRISMA) guidelines (see also the PRISMA-Protocol checklist in online supplemental file 2). A standardised data extraction form (attached as online supplemental file 3) will be used to extract data from the included policy documents and studies. The extracted information will include: study setting; location; year/s; population; participant demographics and baseline characteristics; details of the policies (policy specific to which Article of FCTC or whether beyond FCTC) and comparator conditions; study design; sample size and study completion rates; outcomes and times of measurement; information for assessment of the risk of bias and strength of evidence. Data extraction will be done by two reviewers independently. Missing data will be requested from study authors.

## QUALITY ASSESSMENT
Critical appraisal will be conducted only for academic publications literature, for which we will use the Effective Public Health Practice Project 'Quality Assessment Tool for Quantitative Studies'.[22 23] The tool will assess study quality on eight domains: selection bias, study design, confounders, blinding, data collection methods, withdrawals and drop-outs, intervention integrity, analyses. It also gives an overall global rating of weak/moderate/ strong evidence, which will be used for sensitivity analyses for RQ2.

## DATA SYNTHESIS
For RQ1: Identified policies will be listed according to the focus of the intervention and the target population.

### Focus of intervention

grouped as 'FCTC and non-FCTC' policies.

### Target population

A further grouping by 'children/youth' and 'adults' will be applied.

Identified policies will be described using the Template for Intervention Description and Replication (TiDieR) checklist for completion and understanding of the context; presented in a tabular form.[24] Conclusions will be drawn in relation to the RQ and adequacy of description of the policies across the groupings (tobacco specific and non-specific; children/youth and adults), summarised narratively to highlight gaps in existing policies for ST control.

For RQ 2: Key study characteristics (eg, study design, population, location, risk of bias) will be presented in a tabular form. We will conduct a meta-analysis on the primary outcomes, if: (1) two or more studies are identified for the particular outcome and (2) Heterogeneity in reported effects ($I^2$) is less than 50%[25]

Meta-analyses will be conducted using a fixed-effect model, unless there is evidence of between-study heterogeneity.[26] Where heterogeneity is ≥50% we will explore possible explanations using subgroup analysis, and will not report a pooled estimate of the effect. Heterogeneity (if more than 50% for any specific outcome) will be explored visually using tables and forest plots, by comparing the effect sizes of studies grouped according to potential effect modifiers. These include:

1. Focus of intervention (eg, FCTC or non-FCTC, etc).
2. Target population (eg, children, youth, adults, elderly, etc).
3. Quality of studies.

We will assess the publication bias by using Begg's funnel plot by visual inspection for asymmetry.

Intermediate outcomes (where available for the identified policies) will be described and compared narratively across the FCTC and non-FCTC policies as well as by the target age of the population (children/youth and adults). The ST context and unintended consequences will also be synthesised narratively using a predesigned conceptual framework on the major influences of the tobacco industry, governments and society on ST context (and their interactions); and the interaction between individual factors and ST use context to shape its uptake and consumption.

### CONTEXT

We will extract any context related information from the papers on the following: physical, sociocultural, political (eg, existence of FCTC and years of implementation) and economic aspects. We will use the framework (figure 1), for a narrative synthesis of the contextual elements that are identified from our included studies.

The International Network for Food and Obesity/non-communicable diseases Research, Monitoring and Action

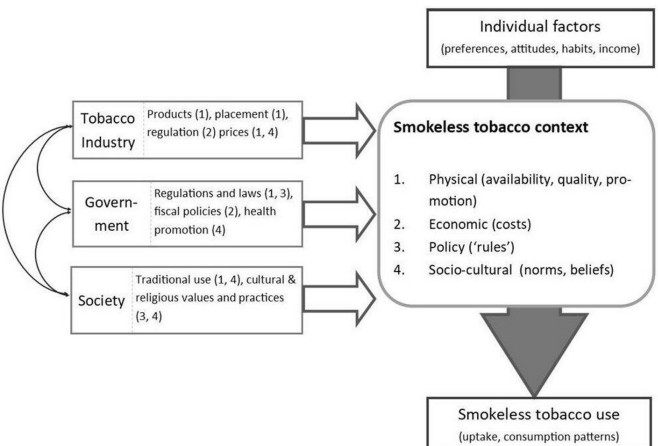

**Figure 1** Contextual framework.

Support (INFORMAS) framework[27] was used for contextualising the basis of this study because it is one of the most frequently used frameworks covering the complex interaction between industry, government and society. It is highly evidence based but equally flexible and can be adapted to other lifestyles such as ST consumption. In addition, it helps to identify the key factors that play a role in explaining the behaviour of individuals and the population and to find ways to modify population behaviour. Although it is not a system map, it can be used as a basis for identifying the main influences. It also allows comparisons to be made between countries and between other areas of research concerned with healthy lifestyles, such as obesity prevention and physical activity. It also allows the transfer of results and helps to identify context-specific and best practice approaches and supports the development and design of policies based on national context.

The ST context framework follows the sociobehavioural theory and a basic premise of this theory is that people learn not only through their own experiences, but also by observing the actions of others and the results of those actions.[28] This is particularly relevant to ST use, which unlike individual smoking behaviour in combustible tobacco forms, is a sociotraditionally influenced behaviour.[29 30] ST consumption is unique in certain regions, such as South Asia, not only because of its traditional heritage and sociocultural myths that promote consumption but also the variety of products from homemade or small scale production to imported license as well as illegal products that are logistically and politically challenging to regulate. Individuals, with their personal factors such as habits, preferences, education and income, interact with the context that determine their use of these products.

The context of ST and each of its component dimensions—physical (eg, availability, quality, promotion), economic (eg, costs, affordability), policy (eg, laws, regulations and accessibility) and sociocultural (eg, norms and beliefs)—has substantial impact on uptake and consumption. Tobacco industry, governments and society can influence and shape these dimensions of

ST context, thus playing a vital role in the uptake and consumption patterns of individuals. These external factors not only interact with each other but also interact with the individuals preferences, attitudes, habits and income to shape their ST use behaviour. For example, governments, at international, national and local levels, through their policies, laws and regulations, provide the 'rules' within which the tobacco manufacturing and sales sector must operate. Through fiscal policies, such as taxation on tobacco sales and subsidies to tobacco growers, governments can influence tobacco product production and prices, and, through health promotion and social marketing, they can also influence sociocultural norms. Society, through its traditional, cultural and religious practices, predominantly establishes the cultural norms for any cultural food practices.

## TIMELINE

The protocol for this review is published on PROSPERO (dated 2 July 2020, V.2).[31] Searches on the databases mentioned in the protocol were conducted during April–May 2020, following which title and abstract screening for scientific literature was done from July to August 2020. We are planning to conduct the full-text screening in October–November 2020 and hold on the data extraction till the protocol is published.

## ETHICS AND DISSEMINATION PLAN

As this study will be a review of published or publicly available data, there are no ethical concerns related to the involvement of humans or any other. Permission for ethics exemption of the review was obtained from the Centre for Chronic Disease Control's Institutional Ethics Committee, India (CCDC_IEC_06_2020; dated 16 April 2020). Data will be collected from publicly available scientific or grey literature through above-mentioned databases. On completion of the review, we will prepare a manuscript for publication in a peer-reviewed journal and present results in national and international conferences.

**Author affiliations**
[1]HRIDAY, New Delhi, India
[2]Health Promotion Division, Public Health Foundation of India, Gurugram, Haryana, India
[3]Department of Health Sciences, University of York, York, UK
[4]School of Public Health, Bielefeld University, Bielefeld, Germany
[5]Ministry of Health, Nutrition and Indigenous Medicine, Colombo, Sri Lanka
[6]Department of Politics, University of York, York, UK
[7]Leibniz Institute for Prevention Research and Epidemiology, Bremen, Germany
[8]University of Dhaka, Dhaka, Bangladesh
[9]Khyber Medical University, Peshawar, Pakistan
[10]Indian Council of Medical Research, India Cancer Research Consortium, New Delhi, India
[11]School of Health, Federation University Australia, Berwick, Victoria, Australia
[12]La Trobe University, Melbourne, Victoria, Australia
[13]Usher Institute, University of Edinburgh, Edinburgh, UK

**Acknowledgements** Ian Kellar (University of Leeds, UK), Subhash Pokhrel (Brunel University, UK), ASTRA Global Health Research Group (https://www.york.ac.uk/healthsciences/research/public-health/projects/astra/).

**Contributors** MA, MM, KS and OD conceptualised the study. MA, AC and NJ drafted the manuscript and MM, MB, SD, SF, RH, ZK, MAR, AR, AV and OD contributed to development of the manuscript as well as finalising the study design. JE provided inputs on data extraction for policy documents, MK reviewed the statistical analysis plan and data extraction plan. RM, KS and AS revised the manuscript critically for intellectual contents. MA and OD supervised the overall process and finalised the manuscript with AC and NJ. All authors approved the final manuscript.

**Funding** This work is funded by the UK's National Institute for Health Research (NIHR) (ASTRA (Grant Reference Number 17/63/76)).

**Disclaimer** The views expressed are those of the authors and not necessarily those of the NIHR or the Department of Health and Social Care.

**Competing interests** None declared.

**Patient consent for publication** Not required.

**Provenance and peer review** Not commissioned; externally peer reviewed.

**ORCID iDs**
Monika Arora http://orcid.org/0000-0001-9987-3933
Aastha Chugh http://orcid.org/0000-0002-6669-3394
Neha Jain http://orcid.org/0000-0002-5449-5980
Melanie Boeckmann http://orcid.org/0000-0001-5909-5508
Sarah Forberger http://orcid.org/0000-0002-7169-675X
Zohaib Khan http://orcid.org/0000-0002-1885-8254
Kamran Siddiqi http://orcid.org/0000-0003-1529-7778

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
