## [Reviewer comments · BMJ Open]

ARTICLE DETAILS

TITLE (PROVISIONAL)	The global impact of tobacco control policies on smokeless tobacco use: a systematic review protocol
AUTHORS	Arora, Monika; Chugh, Aastha; Jain, Neha; Mishu, Masuma; Boeckmann, Melanie; Dahanayake, Suranji; Eckhardt, Jappe; Forberger, Sarah; Huque, Rumana; Kanaan, Mona; Khan, Zohaib; Mehrotra, Ravi; Rahman, Muhammad; Readshaw, Anne; Sheikh, Aziz; Siddiqi, Kamran; Vidyasagaran, Aishwarya; Dogar, Omara

VERSION 1 – REVIEW

REVIEWER	Rizwan S A ICMR - National Institute of Epidemiology, India
REVIEW RETURNED	25-Aug-2020

GENERAL COMMENTS	This protocol addresses an important knowledge gap in the fight against ST. Project timeline dates are not mentioned to assess whether the data collection has been completed or not. Page 11 – line 23 RQ2, provide rationale for these conditions- why 10 studies? And with the wide range of studies in different countries, I square is certainly bound to be on the higher side. Does this mean the authors are essentially precluding any quantitative synthesis? This assumption does not have to be so restrictive. The authors can allow themselves a little room for quantitative synthesis since this might lead to some important hypothesis generation, if not concrete conclusions. Outcomes for RQ2 need more clarity – For e.g., standardized mean difference in quantity consumed, annual percentage change in prevalence, annual decline in specific health outcomes attributable to ST, mean change in attributable number of cases or DALYs lost etc., Also, no statistical considerations for the intended meta-analysis are presented (for e.g, the type of model etc.,). The socio-behavioural theory that underpins the ST context framework (figure 1) needs a short description and the authors must provide a justification for why they think this suits better with adequate references. In databases, what do the authors hope to achieve by searching EMBASE and SCOPUS separately? I understand that the four countries that are selected for detailed document analysis contribute to a majority of the ST burden but
---

	there are some Scandinavian countries where the problem is proportionately dire. I would suggest adding these countries as one unit for analysis. Policies implemented in these countries can provide important pointers to Asian countries. As I understand the field, it is a little ambitious to try to meta-analyse the effectiveness of policy level interventions especially in such a comprehensive but heterogeneous fashion. So I would recommend the authors to aim at one particular intervention and a handful of outcomes in each region. That will narrow down the scope and reduce the heterogeneity of intended results. If some sort of sub-group analysis is planned then it would take care of this. Also, one might encounter situations where a network analysis might be required. Have the authors considered the possibility of encountering ecological correlations between policy implementation and reduction in consumption? If so, how will they handle them in the meta-analysis? Apart from a brief mention of the data extraction form, details of the elements that will be extracted are required.
--	---

REVIEWER	Barbara A. Schillo Truth Initiative
REVIEW RETURNED	31-Aug-2020

GENERAL COMMENTS	This is a very thorough, well written and advanced detailing of a proposed systematic review. I have no major comments. Two minors suggested edits are as follows: Page 5 - references "Ministries" as sections or divisions of govt (Health, Finance, Government). I suggest finding a more generalized terms that applies across countries. Page 7 - line 53 - suggest including English and then regional languages that are included
--

REVIEWER	Michael Iacobelli Johns Hopkins Bloomberg School of Public Health, Baltimore, USA
REVIEW RETURNED	14-Sep-2020

GENERAL COMMENTS	I am pleased to read this protocol paper and its aims at systematically collating smokeless tobacco product policies and regulations globally. The varied approaches by countries that will be identified and cataloged here will be immensely helpful for those of us who conduct research and provide evidence and policy recommendations. Additionally, this study will provide systematic accounting of the effectiveness of smokeless tobacco policies, which is sorely needed. This review provides some comments for clarity. Regarding the FDA's move to consider smokeless tobacco as a modified risk tobacco product. The move wouldn't be an amendment to the law – which would require an act of Congress. But rather it was an authorization made by FDA (under its powers granted by the Family Smoking Prevention and Tobacco Control Act to allow some smokeless tobacco products that were already sold in the USA to be marketed as a modified risk tobacco product. https://www.fda.gov/news-events/press-announcements/fda-grants-first-ever-modified-risk-orders-eight-smokeless-tobacco-products
--

	When assessing RQ1, please clarify if the authors will indicate whether a policy is introduced or passed by the governing body; whether the policy is passed but hasn't been implemented yet due to a lack of implementing rules and regulations; or whether the policy is fully implemented. Additionally, indicate how the authors will manage policy implementation that is stalled or challenged in court by various tobacco industry actors and how that might impact smokeless tobacco control efforts. Please provide as an appendix the standardized data extraction form.
--	--

VERSION 1 – AUTHOR RESPONSE

REVIEWER: 1

This protocol addresses an important knowledge gap in the fight against ST.

Project timeline dates are not mentioned to assess whether the data collection has been completed or not.

Response: Thank you for your comment. The protocol for this review is published on PROSPERO (dated July 2nd 2020, Version 2). Searches on the databases mentioned in the protocol were conducted during April-May 2020, following which title and abstract screening for scientific literature was done from July-August 2020. We are planning to conduct the full-text screening in October-November 2020 and hold on the data extraction till the protocol is published. We have mentioned these details in the manuscript now. (Page 12 Line 24-28)

Page 11 – line 23 RQ2, provide rationale for these conditions- why 10 studies? And with the wide range of studies in different countries, I square is certainly bound to be on the higher side. Does this mean the authors are essentially precluding any quantitative synthesis? This assumption does not have to be so restrictive. The authors can allow themselves a little room for quantitative synthesis since this might lead to some important hypothesis generation, if not concrete conclusions.

Response: Thank you for highlighting this point. We agree that in theory we only need two studies to pool estimates together provided that they are not heterogeneous. We have changed the wording in the protocol to indicate this. If the heterogeneity is higher than the set cut-off, we will not pool the results but instead explore that heterogeneity using sub-group analysis. Groups will be defined by study quality, focus of the intervention and the target population. These are already mentioned in the protocol but we have expanded on the statistical methods under data synthesis now for clarity, see edited text (Page 10 Line 29-32; Page 11 Line 1-10) as follows:

“ For RQ 2: Key study characteristics (e.g., study design, population, location, risk of bias) will be presented in a tabular form. We will conduct a meta-analysis on the primary outcomes, if:

- i) 2 or more studies are identified for the particular outcome, and
- ii) Heterogeneity in reported effects (I-squared) is less than 50%

Meta-analyses will be conducted using a fixed-effect model, unless there is evidence of between-study heterogeneity. Where heterogeneity is $\geq 50\%$ we will explore possible explanations using sub-group analysis, and will not report a pooled estimate of the effect. Heterogeneity (if more than 50% for any specific outcome) will be explored visually using tables and

forest plots, by comparing the effect sizes of studies grouped according to potential effect modifiers. These include:

1. Focus of intervention (e.g. FCTC or non-FCTC, etc.);
2. Target population (e.g. children, youth, adults, elderly, etc.)
3. Quality of studies.

We will assess the publication bias by using Begg's funnel plot by visual inspection for asymmetry.”

Outcomes for RQ2 need more clarity – For e.g., standardized mean difference in quantity consumed, annual percentage change in prevalence, annual decline in specific health outcomes attributable to ST, mean change in attributable number of cases or DALYs lost etc., Also, no statistical considerations for the intended meta-analysis are presented (for e.g., the type of model etc.,).

Response: Thank you for your comments, this has helped us expand on these points. We have now added more clarity to the definition of outcomes and the statistical considerations, as follows in the text (Page 7, Line 11-16):

“Primary outcomes

- percentage change in prevalence of ST use
- rates of quitting ST (proportion of those who stopped out of total intervened)
- percentage change in ST initiation (from never used to ever used)
- change in health outcomes for cancers and cardiovascular diseases attributable to ST (ST-related disease burden is substantial for these conditions, as noted in a recent study)”

The socio-behavioral theory that underpins the ST context framework (figure 1) needs a short description and the authors must provide a justification for why they think this suits better with adequate references.

Response: Thank you for your comment. We have described the framework with some examples to showcase the underpinning theory. We have also provided justification for the use of this model on Page 11, Line 25-33; Page 12 Line 1-22). Please see the following text which has been added to the manuscript.

“ The INFORMAS framework was used for contextualizing the basis of this study because it is one of the most frequently used frameworks covering the complex interaction between industry, government and society. It is highly evidence-based but equally flexible and can be adapted to other lifestyles such as ST consumption. In addition, it helps to identify the key factors that play a role in explaining the behavior of individuals and the population and to find ways to modify population behavior. Although it is not a system map, it can be used as a basis for identifying the main influences. It also allows comparisons to be made between countries and between other areas of research concerned with healthy lifestyles, such as obesity prevention and physical activity. It also allows the transfer of results and helps to identify context-specific and best practice approaches and supports the development and design of policies based on national context.

The ST context framework follows the socio-behavioral theory and a basic premise of this theory is that people learn not only through their own experiences, but also by observing the actions of others and the results of those actions. This is particularly relevant to ST use, which unlike individual smoking behavior in combustible tobacco forms, is a socio-traditionally influenced behavior. ST consumption is unique in certain regions, such as South Asia, not only because of its traditional heritage and socio-cultural myths that promote consumption but also the variety of products from homemade or small scale production to imported licensed as well as illegal products that are

logistically and politically challenging to regulate. Individuals, with their personal factors such as habits, preferences, education and income, interact with the context that determine their use of these products.

The context of ST and each of its component dimensions - physical (e.g. availability, quality, promotion), economic (e.g. costs, affordability), policy (e.g. laws, regulations and accessibility) and socio-cultural (e.g. norms and beliefs) - has substantial impact on uptake and consumption. Tobacco industry, governments and society can influence and shape these dimensions of ST context, thus playing a vital role in the uptake and consumption patterns of individuals. These external factors not only interact with each other but also interact with the individuals' preferences, attitudes, habits and income etc. to shape their ST use behavior. For example, governments, at international, national and local levels, through their policies, laws and regulations, provide the 'rules' within which the tobacco manufacturing and sales sector must operate. Through fiscal policies, such as taxation on tobacco sales and subsidies to tobacco growers, governments can influence tobacco product production and prices, and, through health promotion and social marketing, they can also influence sociocultural norms. Society, through its traditional, cultural and religious practices, predominantly establishes the cultural norms for any cultural food practices."

In databases, what do the authors hope to achieve by searching EMBASE and SCOPUS separately?

Response: We did not realize that these would essentially give us the same papers, therefore both were included for a comprehensive search of the literature. Now we have conducted the searches and progressed through our first stage of screening so it would be difficult to exclude one of these at this stage, but thank you for your suggestion.

I understand that the four countries that are selected for detailed document analysis contribute to a majority of the ST burden but there are some Scandinavian countries where the problem is proportionately dire. I would suggest adding these countries as one unit for analysis. Policies implemented in these countries can provide important pointers to Asian countries.

Response: Thank you for this comment. Only Ministry website searches have been restricted to four countries (Bangladesh, India, Pakistan and Sri Lanka), for which 'language' expertise is available in the research group. The scientific literature and grey literature searches have been inclusive of policies across the globe, including Scandinavian countries to ensure inclusion of all ST related policies globally. We hope this clarifies this query.

As I understand the field, it is a little ambitious to try to meta-analyse the effectiveness of policy level interventions especially in such a comprehensive but heterogeneous fashion. So I would recommend the authors to aim at one particular intervention and a handful of outcomes in each region. That will narrow down the scope and reduce the heterogeneity of intended results. If some sort of sub-group analysis is planned then it would take care of this. Also, one might encounter situations where a network analysis might be required. Have the authors considered the possibility of encountering ecological correlations between policy implementation and reduction in consumption? If so, how will they handle them in the meta-analysis?

Response: Thank you for a detailed and constructive feedback on the proposed meta-analysis. One of the objectives of this review (RQ1) is to collate the policies that have been used for controlling ST and hence, we do expect this exercise to be widely varying in terms of interventions and outcomes. However, we do not intend to restrict this work to any particular intervention or a set of interventions as that will narrow down the scope of this review. We have tried to incorporate your suggestions of exploratory analysis by sub-groups as per earlier comments, which would hopefully take care of this.

We also agree with the reviewer that there is a possibility of encountering ecological correlations between policy implementation and reduction in consumption. We will collect data on whether the analysis in each study was done on individual level or group level. However, we anticipate that there will be a very limited number of articles that will allow us to explore this. In that case we will explore whether these studies differ than the other studies included in our meta-analysis and report our findings accordingly.

Apart from a brief mention of the data extraction form, details of the elements that will be extracted are required.

Response: We have now provided the data extraction form with the manuscript as a supplementary file 3.

REVIEWER: 2

This is a very thorough, well written and advanced detailing of a proposed systematic review. I have no major comments.

Two minors suggested edits are as follows:

Page 5 - references "Ministries" as sections or divisions of govt. (Health, Finance, Government). I suggest finding a more generalized terms that applies across countries.

Response: Thank you for pointing this out. We have now changed it to a more generalized term – 'governmental bodies such as Ministry of Health, Ministry of Finance etc.', as it is relevant to four countries where this search is being done.

Page 7 - line 53 - suggest including English and then regional languages that are included

Response: We have addressed this comment and have included English and then regional languages in the main text in methodology section as below:

"Articles in English and for which linguistic expertise is available within the team (Bengali, Hindi and Urdu) will be included." (Page 5, Line 28)

REVIEWER: 3

I am pleased to read this protocol paper and its aims at systematically collating smokeless tobacco product policies and regulations globally. The varied approaches by countries that will be identified and cataloged here will be immensely helpful for those of us who conduct research and provide evidence and policy recommendations. Additionally, this study will provide systematic accounting of the effectiveness of smokeless tobacco policies, which is sorely needed. This review provides some comments for clarity.

Regarding the FDA's move to consider smokeless tobacco as a modified risk tobacco product. The move wouldn't be an amendment to the law – which would require an act of Congress. But rather it was an authorization made by FDA (under its powers granted by the Family Smoking Prevention and Tobacco Control Act to allow some smokeless tobacco products that were already sold in the USA to

be marketed as a modified risk tobacco product. <https://www.fda.gov/news-events/press-announcements/fda-grants-first-ever-modified-risk-orders-eight-smokeless-tobacco-products>

Response: Thank you for providing this valuable information. We have now revised this in the protocol as follows:

“On the other hand, in the United States of America, the Food and Drug Administration (FDA) now considers ST as a modified risk tobacco product, under their Family Smoking Prevention and Tobacco Control Act 2009”. (Page 5, Line 7-10)

When assessing RQ1, please clarify if the authors will indicate whether a policy is introduced or passed by the governing body; whether the policy is passed but hasn't been implemented yet due to a lack of implementing rules and regulations; or whether the policy is fully implemented. Additionally, indicate how the authors will manage policy implementation that is stalled or challenged in court by various tobacco industry actors and how that might impact smokeless tobacco control efforts.

Response: This is an excellent suggestion by the reviewer. We would have liked to have such a listing but the focus of our systematic review is to list the ST related policies which have been enacted in various countries and to review the impact of these implemented policies on ST outcomes. Therefore, the status of the implementation of the policies, if the policy is applied or how it is enforced cannot be covered at this point. But an excellent suggestion and this could be the basis for another review or extension of this work. However, it would not be feasible to do it under the current review as per the timelines.

Please provide as an appendix the standardized data extraction form.

Response: Thank you for this suggestion. We have now added the data extraction form as a supplementary file 3 to the manuscript.

VERSION 2 – REVIEW

REVIEWER	Rizwan Suliankatchi Abdulkader ICMR- National Institute of Epidemiology, India
REVIEW RETURNED	20-Oct-2020
GENERAL COMMENTS	The replies are satisfactory.